# Engaging citizens in local health policymaking. A realist explorative case-study

Esther De Weger[1,2]*, Hanneke W. Drewes[1], Natascha J. E. Van Vooren[1], Katrien G. Luijkx[2], Caroline A. Baan[2]

1 Centre for Nutrition, Prevent and Health Services, Department of Quality of Care and Health Economics, National Institute for Health and the Environment (RIVM), Bilthoven, The Netherlands, 2 Tilburg University, Tranzo, Tilburg School of Social and Behavioural Sciences, Tilburg, The Netherlands

* esther.de.weger@rivm.nl

## Abstract

### Background

Municipalities have been trying to involve citizens as citizen participation is thought to improve municipalities' accountability, the quality of services, and to align policies and services to communities' needs. This study examined citizens' participation preferences in policymaking by investigating their health policy priorities, expectations of involvement, and required support.

### Methods

For this case-study the realist evaluation approach was applied to focus groups with citizens and to a workshop with a local panel consisting of professionals, citizens and citizen representatives.

### Results

This study showed that citizens want to be involved in (health) policymaking with the aim of improving their communities' quality of life and living environment and prioritised local services and amenities (e.g. suitable housing, public transport, health and care services). Instead, professionals' priorities were focussed on singular public health issues related to prevention and lifestyle factors. The results also show that citizens felt responsible for driving citizen participation and representing community needs to the municipality, but needed the municipality to improve their communication and accessibility in order to do so successfully. Furthermore, the professionals on the panel indicated that they needed training on how to reach out to citizens. Such training should highlight how to better align their language to citizens' lived experiences. They also wanted their organisations to provide more space, flexibility and resources to build relationships with citizens in order to provide improved communication and accessibility to citizens.

### Conclusion

The difference in priorities between citizens and professionals highlights the importance of involving citizens in policymaking. Moreover, citizens' involvement can act as a lever for

**Data Availability Statement:** The qualitative (anonymized) data has been made available in the appendix

**Competing interests:** The authors have declared that no competing interests exist.

change to bring a wider range of services and policy sectors together and has the potential to better align policies to citizens' lived experiences and hopefully increase the democratic legitimacy of policymaking. However, to fulfil such potential municipalities will need to invest in improving their accessibility and communication with communities.

## Introduction

It is widely accepted that the health of a local democracy is dependent on citizens' ability to participate in the life of their communities. Many local governments across the world, therefore, are increasingly trying to engage citizens in the services, policies and decisions which impact their health, lives, and communities [1–3]. More specifically, involving citizens in designing health policies is thought to improve the quality of services and policies and is seen as an effective and meaningful way to ensure health(care) systems are more integrated around communities' needs instead of centred on organisations' traditional remits [4–8]. The premise being that citizens' voices can act as a mechanism for change and can help integrate a wider range of health, care and community services [9–11].

Due to the growing interest in centring health systems more on citizens' needs through citizen participation, municipalities are increasingly applying a range of community engagement (CE) approaches to involve citizens in the decision-making, planning, designing, governance and delivery of health services and policies, including for example, public consultations, peer-delivered health(care) services, citizen's juries and panels, community meetings, and advisory panels [12–14]. CE approaches have been implemented at different participatory levels from e.g. consultation—where people have limited power to influence decision-making—to partnership and (shared) leadership where people have decision-making control [12–15]. In policymaking specifically, local governments have been experimenting with different involvement approaches, including more formal top-down techniques (e.g. The Right to Challenge in the U.K. and The Netherlands) and more informal bottom-up approaches (e.g. community-led initiatives) [16].

Despite a significant evidence-base regarding CE, previous studies have shown that public sector organisations, including municipalities, still struggle to implement CE approaches befitting their own local contexts. For example, some studies have described different approaches for engaging citizens in the policy process and investigated enablers and barriers for the successful implementation of such methods [17–20]. Other studies have examined important factors influencing the effectiveness of such methods including, for example, the scope, credibility and decision-making powers of the approaches [6, 21]. Previous literature has also highlighted how important it is for organisations to enable citizens to be involved according to their own interests. Understanding citizens' interests and priorities helps to align involvement approaches, policies and services to citizens' experiences and needs, rather than to professionals' perspectives and to organisational remits [11]. However, the evidence also shows that organisations especially struggle to be more citizen-centred and find it difficult to take into account people's needs, interests and priorities and thus to empower citizens to be involved on their own terms [22].

Previous literature on citizen involvement in policymaking is largely focussed on cataloguing the different types of engagement approaches in policymaking [9, 19] and on how organisational processes and structures surrounding citizen involvement can be improved [6, 18]. In comparison, there are few studies which focussed on citizens' perspectives and experiences

regarding their involvement in policymaking. This means there is little evidence on citizens' health policy priorities and citizens' involvement preferences. Not enough is known about how citizens themselves would like to be involved in policymaking and what support they feel they require in order to be involved successfully. Nor is there much literature on how organisations and professionals can build better relationships with communities by using citizens' perspectives, experiences, and needs as a starting point.

The aim of this study was to start addressing this knowledge gap by investigating how citizens wished to be involved in developing and implementing local governmental health policies. As such, this case-study investigated citizens' priorities for the local health policy of a rural Dutch municipality. The study examined how citizens wished to be involved in implementing the priorities and the support they required from the municipality in order to be involved in the policy process. This paper presents the results of a realist qualitative case-study, which explored the underlying contextual factors and mechanisms explaining citizens' priorities, their experiences and how they wished to be involved in addressing the priorities. The study, based on findings from focus groups with citizens and on findings from a workshop with a reference panel, explored the following research questions:

- What are citizens' priorities for their municipality's health policy? What are the underlying contextual factors and mechanisms explaining their priorities?

- How do citizens prefer to be involved in addressing the priorities and implementing the local health policy? What are the underlying contextual factors and mechanisms explaining their experiences?

- What support do citizens need in order to be involved in addressing the priorities and implementing the local health policy?

## Methods

This case-study is part of a four-year qualitative multiple case-study evaluating the development of community engagement approaches in six different regions in the Netherlands. The multiple case-study was undertaken in consultation with a reference panel. The panel consisted of stakeholders involved in developing CE approaches within the six regions including policymakers, involved citizens, citizen representatives, and experts in the fields of public health, health inequalities and citizen participation. The panel therefore helped to ensure the study addressed stakeholders' questions regarding community engagement and addressed relevant gaps in the literature. For this case-study, four focus groups with citizens from one of the multiple case-study's municipalities were held. The workshop with the reference panel refined, enriched and validated the focus groups results.

The case-study was informed by the realist evaluation approach. The realist evaluation approach seeks to explain the causal relationships between contexts, mechanisms and outcomes of interest in particular programmes of interest [23]. In this way, the study sought to understand the underlying reasons for citizens' priorities and preferred way to be involved in addressing the priorities *(See Tables 1 and 2 for definitions for realist concepts)*.

### Study sample and data collection for the focus groups

For this case-study, the authors collaborated with a rural Dutch municipality that wanted to better understand how citizens' health and wellbeing is impacted by their broader social environment and to integrate services and policies accordingly. For this study, purposive sampling [30] was used to ensure citizens representing each of the eight villages within the municipality's

**Table 1. Reference panel participant description.**

| Nr. | Type of organization | Type of function |
|---|---|---|
| 1. | Community-led initiative | Volunteer, community-led initiative board member |
| 2. | Community-led initiative | Volunteer, community-led initiative board member |
| 3. | Community-led initiative | Volunteer village key worker, community-led initiative board member |
| 4. | Patient & Public Involvement organization | Representative role, outreach role |
| 5. | Patient & Public Involvement organization | Representative role, project management role |
| 6. | Patient & Public Involvement organization | Representative role, educational role for both citizens & organisations |
| 7. | Patient & Public Involvement organization | Representative role, policymaker |
| 8. | Municipality | Policymaker |
| 9. | Municipality | Policymaker |
| 10. | Municipality | Policymaker |
| 11. | Municipality | Policymaker |
| 12. | Health & care organization | Public health professional |
| 13. | Knowledge institutes | Researcher |
| 14. | Knowledge institutes | Researcher |
| 15. | Knowledge institutes | Researcher |
| 16. | Knowledge institutes | Researcher |
| 17. | Knowledge institutes | Commissioner of research |

boundary were reflected within the sample. Citizens were recruited by the municipality's policymakers' and communication officers' local networks. Citizens were invited by the municipality and were asked to recommend other citizens for the focus group as well (snowball

**Table 2. CE-oriented definitions of realist concepts.**

| | |
|---|---|
| *Intervention* | Refers to interventions' implemented activities, strategies, and resources [24], e.g.: citizen advisory panel meetings, or neighbourhood organized workshops |
| *Context* | Pertains to the backdrop of an intervention and includes the pre-existing organisational structures, cultural norm of the community, the nature and scope of pre-existing networks and geographic location effects [25, 26]. |
| *Mechanism* | Refers to what 'triggers' participants to want to participate or not in an intervention. 'Mechanism' does not refer to the intentional resources offered or strategies implemented within an intervention. Mechanisms usually relate to cognitive, emotional, behavioural responses to intervention resources or strategies [25]. Mechanisms are usually hidden, sensitive to variations in context, and generate outcomes [27]: e.g. citizens feeling more empowered due to learning opportunities. |
| *Outcome* | Refers to intended, unintended, or expected intervention outcomes [25]; e.g. sustainability, quality and integration of services *(macro)*, citizens' level of involvement in health and care services (e.g. in designing policies) *(meso)*, citizens' health and wellbeing outcomes *(micro)*. |
| *CMO* | To understand how certain contextual factors shape or trigger the mechanism, causal links are expressed through 'context-mechanism-outcome configurations' (CMOs). Formulating and refining CMOs is largely how researchers analyse data in RE as it allows for a deeper understanding of which (aspects of) interventions work, for whom, under which circumstances and to what extent [28]. CMOs are also used to generate or refine programme theories, which in turn help shape the final product of an evaluation (e.g. recommendations). CMOs are also to generate or refine programme theories. |

Sources: [11, 22, 29].

sampling). All focus groups were held when and where citizens would normally meet in order to reduce burden on participants.

EdW conducted the focus groups with citizens, while one of the municipality's policy-makers was there to observe and to provide explanations only when required. A semi-structured topic guide was used to anchor the focus group process *(available upon request)*. During the focus groups, citizens were first openly asked to discuss their own priorities for the local health policy. After discussing their own priorities, citizens were asked to discuss the priorities health and care professionals working within the local health system had provided to the municipality a priori and then to rank all priorities (i.e. their own priorities and those of the professionals) in order of importance. Finally, citizens were asked how they wished to be involved in addressing the priorities and the support they required from the municipality.

To aid data analysis, all focus groups were recorded and transcribed. All study participants provided informed consent. The focus group data were supplemented and triangulated by fieldnotes and observations, e.g. of meetings between policymakers and involved citizens, and municipality's council meetings. Furthermore, the focus group findings were anonymised and aggregated, after which the summarised findings were shared with focus group participants for final reflections and feedback to ensure findings had face validity and rigour. Focus groups were conducted until authors agreed the point of data saturation was reached; when no new themes emerged and when there was a high rate of recurrence of responses) [31]. Ultimately, four focus groups (each lasting about two hours) were held with a total of sixteen citizens representing the eight different villages. Finally, the focus group results were shared and discussed during a workshop with the reference panel. The study received ethics approval from Tilburg University (reference EC-2017.96) and data were collected between November 2019 and February 2020.

**Data analysis.** In order to examine citizens' health policy priorities and how they wished to be involved in addressing the priorities and implementing the policy, the authors constructed context-mechanism-outcome configurations (CMOs) within each focus group transcript to understand the contextual factors and mechanisms underlying their priorities and preferred role. Focus groups were thus coded and analysed using CMOs which were drafted and analysed in MaxQDA by EdW, refined and confirmed by NvV and finally discussed by all authors. To aid the authors during the data analysis process and to ensure consistency and transparency, the authors applied the same CE-oriented definitions of 'interventions', 'contexts', 'mechanisms', and 'outcomes' *(See Tables 1 and 2)*. The clustering followed a sequential and iterative process which has been applied in previous studies and described elsewhere [11, 22]:

a. CMOs were coded and clustered into citizens' priorities, their preferred role in addressing these or into municipality's role in addressing the priorities;

b. the authors discussed the clusters and thematically analysed, reviewed and discussed them again.

c. the final draft of the clustered CMOs were shared with all authors to confirm and refine the themes.

d. after the final draft of the CMOs a workshop was held during which the study's initial findings were presented to the reference panel. During the workshop the panel enriched the findings and discussed the validity, relevance, and applicability of the findings within their own local contexts and whether they were experiencing similar issues in the development of their own CE approaches. Confirming that the results had face validity, the workshop provided rich anecdotal evidence, thus further refining the results.

## Results

The following section will first describe the results from the focus groups and highlight citizens' health policy priorities. The way citizens wished to be involved in addressing the priorities and the support citizens required from the municipality to be successfully involved in addressing the priorities will also be addressed in the focus group results. Throughout, citizens' experiences will be highlighted with examples of individual CMOs underpinning the results (*S1 Appendix includes further CMO examples*). The following section will first describe the results from the focus groups, after which the panel's results and reflections will be summarised separately.

### Focus group participants

One focus group involved five citizen, two focus groups involved four citizens, and another one involved three citizens. The average age of interviewed citizens was approximately 45, all interviewees were white Dutch, the majority were women (13 women, three men), most were employed (10), some were retired (4) and others were full-time carers for their spouses (2). All interviewees were volunteers within their villages (16), e.g. at their local football club, church or village council. Citizens were unanimous in the priorities described below, however there were contextual differences in the villages where the citizens lived, which informed how they discussed and ranked the priorities. For example, within one of the focus groups with citizens who lived in a more remote village, they discussed the need to improve the amount and accessibility of services and amenities. Whereas the other three focus groups talked more about the need to maintain the level of services and amenities they currently had access to.

**Reference panel participants.**   Stakeholders from each of the six regions had participated including five citizens, two PPI professionals, two researchers and experts in the field of community engagement, two policymakers, one public health professional, one commissioner of health services research.

### Citizens' health policy priorities stem from their holistic experiences and perceptions and differ from professionals' priorities

The results indicate that citizens' health policy priorities are cross-sectoral and holistic, touching on different policy sectors, including the health and care sector, the housing sector, public transport and the local infrastructure. Citizens perceived and experienced a wide-variety of public issues as having an impact on their health and wellbeing and particularly highlighted the importance of:

a. suitable and affordable housing for local residents, especially for young people (18–25), young families and elderly residents to live independently in their own homes and villages for as long as possible;

b. improving the accessibility of health and care services, i.e. primary care services, general practitioners, dementia care, carers' support;

c. the living environment e.g. public transport and inviting green space to enable all age groups to exercise and to meet socially;

d. local shops and meeting places within village;

e. social activities which can bring different groups of residents together.

The results, as evidenced by the underpinning CMOs, suggest that citizens prioritise services and amenities because these strengthen feelings of social cohesion and ensure villages are

suitable for all age groups, thus helping all citizens to live healthier and happier lives within their villages for as long as possible. Underlying these priorities was the idea that social cohesion was pivotal to citizens' wellbeing and citizens' fear that without such services or amenities local citizens would be forced to move out of the villages to 'bigger city centres' (for young people and families) or to retirement homes away from their families (for elderly residents). For example, one citizen who enjoyed living in one of the villages due to their roots and social network, expressed concern that the recently built housing was unaffordable to local residents, as none of it had been designated as social housing. She felt this was unfair to local residents and worried that due to the housing shortage increasing numbers of local residents would be forced out.

> *"Yeah, just a comfortable village life. . .My family lives here, I grew up here. It's what we are familiar with and we don't want to move away"*

Furthermore, interviewees had also underscored the importance of accessible health and care services. Especially citizens who were carers for elderly relatives perceived that the changes in the Dutch health and care system (e.g. decentralisation, Participation Act 2015) now (over)emphasised citizens' independence in seeking out and arranging suitable care for their older family members *(context)*. They felt that carers were now under too much pressure especially because they experienced the current health and care system as fragmented and difficult to navigate *(mechanism)*. They wanted better and more accessible carers' support and better signposting of available services *(outcome)*, in part as this would help to reduce carers' stress and isolation *(outcome)*. Some citizens enjoyed living in one of the villages as it provided them with easy access to primary care services, stating it 'made life easier' *(See S1 Appendix for more CMO examples regarding residents' priorities)*.

When citizens were presented with professionals' priorities (after first openly discussing their own priorities) it became clear that residents and professionals perceived the priorities for the local health policy differently and citizens mostly ranked professionals' priorities below their own. Professionals had highlighted public health topics such as prevention and lifestyle (e.g. a reduction in obesity, smoking and alcohol rates and an improvement in healthy eating and exercise rates). While citizens understood why professionals had highlighted public health issues, they did not see these as priorities and were not concerned with obesity or alcohol rates. Instead, citizens prioritised services, public (green) spaces, and social activities which would support their own health and wellbeing and their social connection to other citizens. However, there was some overlap between citizens and professionals when it came to dementia care and carers' support. Both saw these as important priorities and described how a lack of support increased stress levels and social isolation for those involved.

**Reference panel reflections.**   The panel validated the focus group results and acknowledged that citizens' health policy priorities were more cross-sectoral, holistic and based on their own lived experiences, rather than on healthcare systems' remits. However, the professionals on the panel also felt that citizens and professionals often overlap in their health policy priorities, but that they label and approach these priorities differently. The professionals used the example that citizens prioritise 'social cohesion' and want to organise more social activities, while professionals discuss loneliness and want better carers' support. Thus the professionals within the panel felt that the gap between citizens and the health and care system is not as big as often assumed and can be bridged by aligning health policies' language better to citizens' vocabulary and framing.

## Citizens felt ownership for enabling citizens' engagement within the community by organising social activities and feeding back citizens' wider health and care needs and experiences to the municipality

The following theme highlights that citizens felt ownership for enabling other citizens to also be involved in the community by organising social activities. They also felt responsible for sharing citizens' experiences and needs with the municipality in order to improve local health (care) services and policies and the living environment. All interviewees were actively involved in community-led initiatives, including e.g. local football clubs, church parishes, village councils to improve village life *(context)*. While they all held different roles in different types of community-led initiatives they all shared a sense of community and reciprocity *(mechanism)*. They also stated that being engaged in such a manner helped them to maintain or improve their own social networks *(outcomes)* and helped them to improve the social cohesion within the villages *(outcome)*. Additionally, citizens involved in church councils, village councils, and the municipality's Advisory Board felt responsible for sharing citizens' experiences, concerns, and ideas regarding local services, amenities, and the living environment with the municipality and wanted to play a linking-pin role in ensuring citizens' perspectives were included and reflected within municipality's plans and policies. They felt that reflecting back people's experiences and needs would help to improve the quality of health(care) services and the living environment as these would be more centred on citizens' priorities and experiences *(See S1 Appendix for CMO examples)*.

**Reference panel reflections.** The panel validated these findings and recognised that citizens feel a responsibility for building and improving their communities. They also highlighted that citizens have a more practical approach to building a healthy and socially active community. Comparatively, professionals tend to stick more to their own organisational remits and national health policy topics. The panel felt these different approaches to local health policies were complementary. They suggested that organisations can leverage these different approaches to develop more holistic policies which are better aligned with citizens' needs as long as organisations are more transparent and accessible to citizens. Both professionals and citizens on the panel mentioned organisations should improve their communication and align the language within policies better with the way citizens experience and describe health related issues.

## Citizens need the municipality to better facilitate their involvement by improving their communication and accessibility for residents and community-led initiatives

Citizens had also discussed the barriers they experienced when organising social activities and collaborating with the municipality and how they wished the municipality would better support them in their involvement. Firstly, interviewees described how engaged residents were experiencing pressure due to increasingly higher workloads. This partly stemmed from the fact that the number of residents willing to volunteer was declining, but volunteers also felt that they had taken on tasks beyond their own remits which were perceived to have been traditionally within the municipality's remit. For example, one participant described how they were now supporting low-income members to navigate the municipal care system and helping them to fill out the right forms and not only so they could continue paying their membership fees *(context)*. This made the volunteer feel increasingly frustrated with their role as it made it more 'bureaucratic' and less fun *(mechanism)*. They worried that this expanding volunteer-job remit, and the reduction in the number of volunteers, meant that it was becoming harder to keep the club going as fewer citizens would want to join or help out *(outcome)*. Secondly,

residents also described how it was more difficult to organise new activities or to further develop and improve their community-led initiatives as the municipality did not clearly communicate (new) funding opportunities or because it required initiatives to increasingly follow bureaucratic rules when applying for new funding or social activities. Finally, interviewees recognised that each community-led initiative only organised activities for their own 'usual suspects' and therefore did not offer opportunities for social bridging (across e.g. different age groups, religious and non-religious residents).

To overcome these barriers, interviewees said that one of the most important things the municipality could do was to listen and improve their communication with citizens. They suggested the municipality improve their accessibility by 'getting out of the municipality building' and attending meetings and social activities organised by citizens instead. They felt this would show a more open-approach and a willingness to listen to citizens. For example, citizens specifically mentioned it would help initiatives to have one central point of contact at the municipality and that better communication and outreach regarding services and funding opportunities for community-led initiatives would facilitate citizens' involvement and volunteering. They also suggested that the municipality should help initiatives to reach a wider audience (than the 'usual suspects') by promoting the social activities and volunteering opportunities more widely *(See S1 Appendix for CMO examples)*.

*"I've noticed how important it is to feel heard as a citizen"*

**Reference panel reflection.** The panel validated the focus groups' findings and stated that the healthcare system should be better aligned with citizens' perspectives, experiences and needs. The professionals on the panel stated that they were still searching for ways to leverage village councils' and community-led initiatives' energy, skills and knowledge with organisations' capability and resources. However, professionals also stated that they needed training on how to reach out to citizens and to better align their language and interventions to citizens' lived experience. Professionals also wanted their organisations to provide them more space, flexibility, and resources to build relationships with citizens and ultimately to align the health and care system to communities' lived experience. Relatedly, the citizens on the panel stated that they would like organisations to invest financially in their initiatives and engagement activities and wanted organisations and professionals to more proactively include citizens in community development and policymaking; they stated that such engagement and outreach from organisations was still too sporadic.

## Discussion

Using the realist evaluation approach, this explorative case-study investigated citizens' priorities for their municipality's local health policy, how they wished to be involved in addressing these priorities and the support they needed from the municipality to be involved.

The study indicates that in rural areas where citizens perceive the local public infrastructure and their village way of life to be under pressure, they prioritise services, amenities, and a living environment which empower them to improve their health and wellbeing and enable them to feel connected to their community. This stands in comparison with professionals who prioritise singular public health issues concerning prevention and lifestyle factors. Citizens will likely not prioritise such lifestyle factors until their more primary needs (i.e. health, wellbeing, and social connectivity) are met first. Furthermore, by comparing citizens' and professionals' priorities, the study underscores the value of involving citizens in the policymaking process. The fact that citizens' priorities were cross-sectoral and focussed on citizens' overall quality of life,

while professionals' priorities were mostly focussed on singular public health topics (e.g. smoking, obesity rates) highlights that citizens' involvement can indeed act as a lever for change to bring a wider range of services and policy sectors together and can thereby help to centre health systems based on citizens' holistic needs [11, 32, 33]. Though the panel's discussions suggest that professionals may minimise this gap between citizens and organisations as they viewed such differences in priorities as based on differences in language and framing (rather than deep rooted differences) which can be (partly) bridged by aligning policies' languages better to citizens' vocabulary and framing.

The study has also highlighted that citizens want to help address health policy priorities and feel ownership for enabling others to be connected and involved within the community by organising social activities and by reflecting other citizens' health(care) experiences and needs back to the municipality. However, within the context of decreasing numbers of volunteers as described above, the decentralisation of health and care services in the Netherlands (e.g. the Participation Act 2015; Living Environment Act 2021), and the corresponding pressure on current volunteers to take on jobs that extend beyond their remits, it raises interesting questions regarding the wider role citizens might be able to play in developing and implementing policies and services to improve communities' health and wellbeing and their living environments [22]. While this municipality was comfortable in encouraging citizens in the relatively demarcated role described above, other regions in, for example the U.K and the Netherlands, are exploring to what extent the development, implementation, and governance of services can be devolved to residents and communities themselves. A clear example of this search for devolvement can be found in the Right to Challenge and the UK Localism Act (2011) [34–36]. The "Right to Challenge" whereby citizens can take over any type of municipal service (e.g. upkeep of the living environment, 'low-level' health(care) services) as long as they can prove that the quality and cost-effectiveness would be improved, shows that citizens' role in addressing health policies have the potential to be extended significantly further. However, the extent to which their roles can be successfully extended will depend on a variety of contextual factors, e.g. local population's interest in doing so, the accessibility and leadership of policymakers and local councillors for citizens, and the extent to which a municipality is willing to experiment with new forms of governance and to contribute financially [34]. Relatedly, the above shows the increasing role municipalities can play in integrating citizen-centred health systems, especially within the context of decentralisation and devolvement to communities, e.g. Manchester Devolution [37].

Furthermore, the study has highlighted ways in which municipalities can support citizens when involved in the policy process to ensure health systems are better integrated based on communities' needs. It underscored the importance of providing facilitative leadership as the citizens clearly described how the municipality's lack of communication and accessibility hindered their involvement [11, 22, 38, 39]. They also highlighted that earlier involvement in the policy process helped them feel more engaged and like their input and ideas mattered [8, 11, 40–43]. Furthermore, citizens stated that they felt a responsibility in enabling other citizens' engagement, thus strengthening social cohesion (a key priority), but that they required more support from the municipality in reaching out to not-yet-engaged citizens as this would increase social bridging. By having a better cross-section of citizens involved in community-led initiatives, they could potentially provide more representative input and feedback to the municipality, which in turn could support the municipality to develop and implement better quality services and policies. At the same time, the local reference panel highlighted the importance of training professionals to provide such support to citizens and wanted more flexibility, space and resources to reach out to citizens and to build relationships with communities.

Finally, it is worth noting that both the involved citizens and policymakers had reported they found citizens' involvement to be of value. During a meeting between local councillors and policymakers, councillors had stated it had enabled the municipality to create a health policy more aligned with citizens' own needs and interests and had helped them to bring different policy sectors together (including health, public transport, environment sectors). While citizens felt like they had established a better connection with their municipality and felt more heard. These may not be hard (health) outcomes (yet), but ought perhaps not be underestimated for their potential value in strengthening local democracy. For example, previously Thurston [44] had suggested that citizens' involvement should not (exclusively) be measured by whether governance decisions were effectively made, but also whether new partnerships were established which could inform health policy and challenge the 'status quo' and add health priorities to the agenda.

## Study limitations

Study limitations included the fact that this case-study only examined one rural municipality's approach to involving citizens in policymaking. Presumably, the rural context of this case-study influenced several important factors in the policy process, including e.g. citizens' priorities, the extent to which citizens are already involved by the municipality, the scope policymakers are given to innovate and include citizens, the extent to which local counsellors listen to the policymakers and by extension citizens. Relatedly, while participants of this case-study covered all villages within the municipality's boundary and covered a range of ages and employment and educational statuses, all participants were already involved within their own communities in some shape or form which meant they already had some links with the municipality. Citizens not already involved in their own communities are likely to have similar policy priorities, but may feel different about their roles in addressing the priorities. However, this case-study was, as far as the authors are aware, a first attempt to show municipalities and health(care) organisations how they can align their involvement approaches, services, and policies to citizens' needs and priorities. Finally, there was a concern that the active participation of one of the municipality's policymakers during the focus groups would prevent citizens from speaking openly about their experiences and collaboration with the municipality. However, in practice, citizens seemed to appreciate the opportunity to share their experiences not just with the researchers but also directly with the policymaker and that this had in fact helped them to feel more heard by the municipality.

The study limitations were mitigated by the reference panel's workshop discussions as this validated and enriched the focus group findings. The panel suggested that the focus group results were representative. of the wider experiences regarding the development and implementation of CE in the Netherlands.

## Future studies

This case-study seems to indicate that citizen involvement can help to better aligns policies to citizens' lived experiences and could therefore increase the democratic legitimacy of policymaking. However, future studies are required, including (sub)urban municipalities and different citizens (e.g. harder-to-reach groups like, low-income households, elderly residents) to explore this further. For example, future research could investigate whether the gap between citizens and organisations is based on differences in language and framing, as the professionals on the local reference panel presumed and/or is rooted in more systemic issues, e.g. lack of available resources and collaborative working with citizens beyond organisational remits. Ultimately, future studies should also investigate if and how citizens' involvement actually results

in different policies and if and how the content of policies is indeed better aligned with citizens' needs and lived experiences.

## Conclusion

This study investigated how citizens preferred to be involved in policymaking by examining citizens' health policy priorities, how citizens wished to be involved in addressing the priorities, and the support they required. The study shows that citizens in rural areas who perceive the local public infrastructure to be under pressure prioritise means that help improve their quality of life and living environment and help them to feel more connected to their communities, e.g. affordable and suitable housing, accessible health(care) services, and social activities. By contrast, professionals focus on singular public health issues related to prevention and lifestyle factors. This difference in priorities highlights the importance of involving citizens in policymaking and how citizens' involvement can act as a lever for change to bring a wider range of services and policy sectors together. Citizens' involvement has the potential to better align policies to citizens' lived experiences and hopefully to increase the democratic legitimacy of policymaking. However, to fulfil such potential municipalities will need to prioritise community engagement and invest in improving their accessibility and communication for communities.

## Supporting information

**S1 Appendix. Summary of CMOs.**
(DOCX)

## Acknowledgments

The authors would like to acknowledge and thank the residents, the municipality (especially Zoe Cremers) for their openness, collaboration and valuable insights. The authors also acknowledge the policymakers' sincere belief in local government.

## Author Contributions

**Conceptualization:** Esther De Weger, Hanneke W. Drewes, Katrien G. Luijkx, Caroline A. Baan.

**Data curation:** Esther De Weger.

**Formal analysis:** Esther De Weger, Hanneke W. Drewes, Natascha J. E. Van Vooren.

**Investigation:** Hanneke W. Drewes.

**Methodology:** Esther De Weger, Hanneke W. Drewes, Natascha J. E. Van Vooren.

**Supervision:** Hanneke W. Drewes, Caroline A. Baan.

**Writing – original draft:** Esther De Weger.

**Writing – review & editing:** Esther De Weger, Hanneke W. Drewes, Katrien G. Luijkx, Caroline A. Baan.

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
