## [Decision Letter · Decision Letter 0]

22 Dec 2021

PONE-D-21-20040

Engaging Citizens in Local Health Policy-making. A  Realist Explorative Case-Study.

PLOS ONE

Dear Dr. de Weger

Thank you for submitting your manuscript to PLOS ONE. After careful consideration, we feel that it has merit but does not fully meet PLOS ONE’s publication criteria as it currently stands. Therefore, we invite you to submit a revised version of the manuscript that addresses the points raised during the review process.

A rebuttal letter that responds to each point raised by the academic editor and reviewer(s). You should upload this letter as a separate file labeled 'Response to Reviewers'.A marked-up copy of your manuscript that highlights changes made to the original version. You should upload this as a separate file labeled 'Revised Manuscript with Track Changes'.An unmarked version of your revised paper without tracked changes. You should upload this as a separate file labeled 'Manuscript

We look forward to receiving your revised manuscript.

Kind regards,

Yogan Pillay, Phd

Academic Editor

PLOS ONE

Journal Requirements:

Reviewers' comments:

Reviewer's Responses to Questions

**Comments to the Author**

1. Is the manuscript technically sound, and do the data support the conclusions?

Reviewer #1: Yes

2. Has the statistical analysis been performed appropriately and rigorously? 

Reviewer #1: N/A

3. Have the authors made all data underlying the findings in their manuscript fully available?

Reviewer #1: Yes

4. Is the manuscript presented in an intelligible fashion and written in standard English?

Reviewer #1: Yes

5. Review Comments to the Author

Reviewer #1: This is an interesting, well written study of citizen engagement in rural Netherlands. The methodology (realist evaluation) is well executed and presented.

My main comments are as follows:

- it took me until page 11 to confirm that this study was conducted in a rural Dutch municipality (even if the authors and their affiliations provided clues earlier). As such, the research comes across as oddly decontextualised for a realist evaluation that explicitly treats phenomena as inherently context bound. I recommend signalling study location in the title or at minimum in the abstract. Similarly, in the introduction, the literature appears to reference experiences in western democracies - but the way it is currently presented suggests a universality beyond this setting;

- the abstract stays at a descriptive level and does not adequately convey the deeper (and more interesting) analysis provided in the findings section through the CMO configurations. There is nothing here to suggest that a realist analysis was indeed conducted. I recommend reporting something of the CMOs analysis in the abstract;

- in the discussion/conclusion the authors seem to propose that better communication and community engagement is the "lever for change to bring a wider range of services and policy sectors together". However they do not consider the barriers to this outside of a research context where the mediation of researchers is able to counter the dominance of professional world views. Nor do the authors consider how the devolution of care functions to volunteers and households is a well established element of wider (neoliberal) macro-economic logics. The conclusions thus risk coming across as somewhat apolitical and naive.

Minor typos

Last line of Table 1 - the word 'used' needs to be added before 'to generate'

Page 17, para two, first sentence - the word 'had' in the middle of the sentence could be removed; similarly, on page 19, in the last para, first line

Page 23, first line, needs to be rephrased to make the meaning clearer (change 'organisations'' to 'organisational'?)

6. PLOS authors have the option to publish the peer review history of their article (what does this mean?). If published, this will include your full peer review and any attached files.

Reviewer #1: No

---

## [Author Response · Author response to Decision Letter 0]

1 Feb 2022

Reviewer comment 1:

1. Minor typos

Last line of Table 1 - the word 'used' needs to be added before 'to generate'

Page 17, para two, first sentence - the word 'had' in the middle of the sentence could be removed; similarly, on page 19, in the last para, first line

Page 23, first line, needs to be rephrased to make the meaning clearer (change 'organisations'' to 'organisational'?) 

Author response to reviewer comment 1:

We would like to thank the reviewer pointing out the typo’s and the opportunity to correct these. We have made the required changes.

Reviewer comment 2:

2. My main comments are as follows:

- it took me until page 11 to confirm that this study was conducted in a rural Dutch municipality (even if the authors and their affiliations provided clues earlier). As such, the research comes across as oddly decontextualised for a realist evaluation that explicitly treats phenomena as inherently context bound. I recommend signalling study location in the title or at minimum in the abstract. Similarly, in the introduction, the literature appears to reference experiences in western democracies - but the way it is currently presented suggests a universality beyond this setting.

Authors response to reviewer comment 2: 

We agree with the reviewer that the context of the municipality is important and are therefore happy to provide more information. We have adapted the text as shown below and have also added a new table with more detailed description of the municipality including access to public services and the living environment (see Table 3). Furthermore, we have also added a new table (See Table 2) describing the participants from the reference panel to better highlight that we have tested the findings of this study with a wider range of participants from different parts of the Netherlands, providing more contextual information on both the municipality and the reference panel. 

Adapted text in the title:

Engaging Citizens in Local Health Policymaking in a Dutch Municipality. A Realist Explorative Case-Study (Line 1, page 1)

Adapted text in the abstract: 

• Methods: For this case-study the realist evaluation approach was applied to focus groups with citizens of a Dutch municipality. The municipality is made up of several different, smaller villages, some were rural and others suburban and each had their own sense of identity. These results were then presented during a workshop to a reference panel consisting of professionals, citizens, and citizen representatives from across the Netherlands. The panel enriched and refined the results. (Lines 25-30, page 2).

Adapted text in the methods section: 

• For this case-study, the authors collaborated with a Dutch municipality that wanted to better understand how citizens’ health and wellbeing is impacted by their broader social environment and to integrate services and policies accordingly. The municipality was made up of several different villages, some of which were more rural and others more suburban. Each village had its own and distinct sense of identity and within some of the villages Protestant churches played a more prominent role in the social cohesion of the corresponding community (See Table 3 for more information on the municipality. (Pages 9: Lines: 155-161)

Reviewer comment 3:

3. The abstract stays at a descriptive level and does not adequately convey the deeper (and more interesting) analysis provided in the findings section through the CMO configurations. There is nothing here to suggest that a realist analysis was indeed conducted. I recommend reporting something of the CMOs analysis in the abstract. We agree with the reviewer and appreciate the opportunity to improve our abstract accordingly.

Author response to reviewer comment 3:

Adapted text within the abstract:

• Methods: For this case-study the realist evaluation approach was applied to focus groups with citizens of a Dutch municipality. The municipality is made up of several different, smaller villages, some were rural and others suburban and each had their own sense of identity. These results were then presented during a workshop to a reference panel consisting of professionals, citizens, and citizen representatives from across the Netherlands. The panel added to and refined the results. (Lines 25-30).

• Within smaller communities, citizens prioritised local services and amenities (e.g. suitable housing, public transport, health and care services), because these were thought to strengthen feelings of social cohesion and were important to ensuring villages are suitable for all age groups. Underlying these priorities was the fear that without such services or amenities local citizens would be forced to move out of their villages (Lines: 29-34, page 2)

• The results also show that citizens had a sense of community and reciprocity and felt responsible for driving citizen participation and representing community needs to the municipality (Lines: 35-37, Page 2). 

Reviewer comment 4:

4. In the discussion/conclusion the authors seem to propose that better communication and community engagement is the "lever for change to bring a wider range of services and policy sectors together". However they do not consider the barriers to this outside of a research context where the mediation of researchers is able to counter the dominance of professional world views. Nor do the authors consider how the devolution of care functions to volunteers and households is a well-established element of wider (neoliberal) macro-economic logics. The conclusions thus risk coming across as somewhat apolitical and naive. 

Authors reponse to reviewer comment 4:

The reviewer brings up two interesting and important points. The first one concerns the more active role researchers can and should take to help reduce the barriers citizens experience and to ensure citizens’ perspectives, experiences and needs are better represented within health and care systems. To address the first point we have included within the discussion a reflection, based on our own experiences within this study, on how researchers themselves can help to ensure citizens’ voices are more included in policymaking by reducing inaccessibility related barriers. 

The second point concerns the impact that the decentralisation of health and care services has on community engagement and on volunteers and carers. Such discussions also take place in the Netherlands as previous rapports have questioned the motivations behind the decentralisation while at the same time also implementing national policies to increase community engagement any fear such policies have been implemented not to improve the person-centredness of policies and services or to improve the legitimacy of local democracy but to reduce health and care costs by placing the burden on communities and carers themselves (Kleider et al 2018; Pierson 1995; Zwaan 2016). Indeed volunteers and carers within this study had experienced the pressure of higher workloads. While (neoliberal) ideologies regarding (the devolution of) health care systems was not within the scope of our research questions, we have added our own reflections on this topic within the discussion, suggesting that community engagement within a decentralised context needs to be accompanied with sufficient investment in communities to take on these additional roles (Discussion: Lines 436-445). 

Finally, we have included more information on the barriers experienced by citizens’ during their involvement and made suggestions for how these can be addressed. 

Adapted text within abstract:

Conclusion: The differences in citizens’ and professionals’ policy priorities highlights the importance of involving citizens in policymaking. It highlights that citizens have valuable insights and ideas as to how policies can be improved and can be better aligned to citizens’ lived experiences. However, to fulfil such potential municipalities will need to invest in the engagement environment and provide more facilitative leadership to citizens. This will enable citizen involvement to be properly embedded with organizational cultures. This study highlights that improving municipalities’ accessibility and communication with communities is an important part of embedding citizen involvement within organizational cultures as it helps municipalities to truly listen to citizens’ input (instead of minimizing the differences between professionals’ and citizens’ perspectives). Without such changes in municipalities’ cultures and investments in the engagement environment, citizens’ input and roles in policymaking will likely remain more limited. (Page 3, lines 47-58).

Adapted text within the discussion: 

• The fact that organisations may attempt to minimise the differences between citizens’ and professionals’ perspectives highlights why researchers, through more participatory approaches, should advocate for citizens’ priorities and needs and suggest ways to ensure that especially ‘unwanted voices’—e.g. loud voices, dissenting voices, or those of the often dismissed ‘usual suspects’—are included as a key part of local health policies (Beresford 2019). In this way, researchers can help to bridge the gap between citizens and professionals that citizens often experience (e.g. due to differences in language or power imbalances) and to improve municipalities’ communication and accessibility for citizens (Beresford 2019; De Weger et al 2020 & 2022). (Page 20, Lines: 406-416).

• However, previous literature has shown that such a devolvement requires not just the devolvement of (some) tasks and responsibilities to communities, but also for national and local governments to cede some power and resources (including information and funds) to communities to conduct these tasks and responsibilities successfully (Abimbola et al 2019). Without such devolvement and corresponding investments in the engagement environment, CE risks increasing the burden on engaged citizens and carers as some of this study’s participants suggested. In line with previous literature, this study suggests that for successful citizen involvement, communities and municipalities should transparently discuss what roles and tasks communities can and want to fulfil and which should remain with municipalities (De Weger et al 2018 & 2020). (Page 22, Lines: 443-453).

• At the same time, the reference panel highlighted the importance of training professionals to provide such support to citizens and wanted more flexibility, space and resources to reach out to citizens and to build relationships with communities. Furthermore, previous literature has shown that besides such training and resources, municipalities will need to improve the engagement environment by, for example, addressing experienced power imbalances between communities and professionals, and providing the leadership to actually use citizens’ input to meaningfully change and implement policies accordingly (Beresford 2019; De Weger et al 2018 & 2020; Holley 2016). (Page 23, Lines: 471-475). 

Adapted text within conclusion:

• This difference in priorities highlights the importance of involving citizens in policymaking as such involvement has the potential to improve policymaking and to better align policies to citizens’ lived experiences. However, to fulfil this potential municipalities will need to invest in the engagement environment and provide more facilitative leadership to citizens. This will help to embed citizen involvement within organisational cultures. This study shows that improving municipalities’ accessibility and communications with communities is an important part of embedding citizen involvement within organisational cultures as it helps municipalities to truly listen to citizens’ input (instead of minimising the differences between citizens’ and professionals’ perspectives). Without such changes in municipalities’ cultures and investments in the engagement environment, citizens’ input and roles in policymaking will likely remain more limited (Pages 26 Lines: 533-542).

---

## [Editor Report · Decision Letter 1]

2 Mar 2022

Engaging Citizens in Local Health Policymaking in a Dutch Municipality. A Realist Explorative Case-Study

PONE-D-21-20040R1

Dear Ms de Weger,

We’re pleased to inform you that your manuscript has been judged scientifically suitable for publication and will be formally accepted for publication once it meets all outstanding technical requirements.

Kind regards,

Yogan Pillay, Phd

Academic Editor

PLOS ONE
---

## [Editor Report · Acceptance letter]

11 Mar 2022

PONE-D-21-20040R1 

Engaging Citizens in Local Health Policymaking. A  Realist Explorative Case-Study 

Dear Dr. De Weger :

I'm pleased to inform you that your manuscript has been deemed suitable for publication in PLOS ONE. Congratulations! Your manuscript is now with our production department. 

Kind regards, 

on behalf of

Dr. Yogan Pillay 

Academic Editor

PLOS ONE